# Faecal Volatile Organic Compound Analysis in De Novo Paediatric Inflammatory Bowel Disease by Gas Chromatography–Ion Mobility Spectrometry: A Case–Control Study

**DOI:** 10.3390/s24092727

**Published:** 2024-04-25

**Authors:** Eva Vermeer, Jasmijn Z. Jagt, Trenton K. Stewart, James A. Covington, Eduard A. Struys, Robert de Jonge, Nanne K. H. de Boer, Tim G. J. de Meij

**Affiliations:** 1Department of Paediatric Gastroenterology, Emma Children’s Hospital, Amsterdam University Medical Centre, 1105 AZ Amsterdam, The Netherlands; j.jagt1@amsterdamumc.nl (J.Z.J.); t.demeij@amsterdamumc.nl (T.G.J.d.M.); 2Amsterdam Gastroenterology Endocrinology Metabolism (AGEM) Research Institute, Amsterdam University Medical Centre, 1105 AZ Amsterdam, The Netherlands; khn.deboer@amsterdamumc.nl; 3Amsterdam Reproduction & Development (AR&D) Research Institute, Amsterdam University Medical Centre, 1105 AZ Amsterdam, The Netherlands; 4School of Engineering, University of Warwick, Coventry CV4 7AL, UK; trenton.stewart@warwick.ac.uk (T.K.S.); j.a.covington@warwick.ac.uk (J.A.C.); 5Department of Laboratory Medicine, Amsterdam University Medical Centre, 1105 AZ Amsterdam, The Netherlands; e.struys@amsterdamumc.nl (E.A.S.); r.dejonge1@amsterdamumc.nl (R.d.J.); 6Department of Gastroenterology and Hepatology, Amsterdam University Medical Centre, 1105 AZ Amsterdam, The Netherlands

**Keywords:** paediatric inflammatory bowel disease, faecal volatile organic compounds, gas chromatography–ion mobility spectrometry

## Abstract

The gut microbiota and its related metabolites differ between inflammatory bowel disease (IBD) patients and healthy controls. In this study, we compared faecal volatile organic compound (VOC) patterns of paediatric IBD patients and controls with gastrointestinal symptoms (CGIs). Additionally, we aimed to assess if baseline VOC profiles could predict treatment response in paediatric IBD patients. We collected faecal samples from a cohort of de novo therapy-naïve paediatric IBD patients and CGIs. VOCs were analysed using gas chromatography–ion mobility spectrometry (GC-IMS). Response was defined as a combination of clinical response based on disease activity scores, without requiring treatment escalation. We included 109 paediatric IBD patients and 75 CGIs, aged 4 to 17 years. Faecal VOC profiles of paediatric IBD patients were distinguishable from those of CGIs (AUC ± 95% CI, *p*-values: 0.71 (0.64–0.79), <0.001). This discrimination was observed in both Crohn’s disease (CD) (0.75 (0.67–0.84), <0.001) and ulcerative colitis (UC) (0.67 (0.56–0.78), 0.01) patients. VOC profiles between CD and UC patients were not distinguishable (0.57 (0.45–0.69), 0.87). Baseline VOC profiles of responders did not differ from non-responders (0.70 (0.58–0.83), 0.1). In conclusion, faecal VOC profiles of paediatric IBD patients differ significantly from those of CGIs.

## 1. Introduction

Inflammatory bowel disease (IBD) is a chronic condition characterised by recurring inflammation of the gut, which can be further categorised as Crohn’s disease (CD), ulcerative colitis (UC), or IBD-unclassified (IBD-U). The past few decades, the prevalence of IBD has been rising, especially in the paediatric population [1]. In parts of Europe, the incidence of paediatric CD has increased up to 9–10 cases per 100,000 individuals [2]. The reasons for this increase are still unknown due to the lack of understanding of the pathogenesis of IBD. It is widely accepted that this disease stems from a complex interplay between host genetics, immune responses, environmental factors, and the intestinal microbiome, alongside its related metabolites [3,4]. 

In recent years, the study of metabolomics, including volatile organic compounds (VOCs), has gained interest due to the potential of VOCs to serve as non-invasive biomarkers in the diagnostic process and monitoring of IBD [5,6]. VOCs are carbon-based molecules and are considered to reflect intestinal microbial composition and function [7]. In adults, faecal VOCs have been shown to differ between IBD patients and controls with gastrointestinal symptoms (CGIs) and between active and inactive IBD [8,9]. In paediatric IBD, our research group has found similar results, with faecal VOCs being able to distinguish IBD patients from healthy controls [10]. In a proof of principle study, we previously showed that paediatric IBD patients have different VOC compositions than children with functional gastrointestinal disorders using field asymmetric ion mobility spectrometry (FAIMS) [11]. 

There is a wide range of approaches that can be used to detect chemical components in a faecal headspace. The recognised gold standard for this is the use of gas chromatography–mass spectrometry (GC-MS), which separates complex chemicals in headspaces through the combination of a pre-separation technique followed by mass analysis [12]. Though this offers high sensitivity and specificity, they require specialised staff and support infrastructure. An alternative is the use of electronic noses. First developed in the 1980s, these instruments use an array of discrete chemical sensors, where headspace components are analysed as a whole with pre-separation [13]. In this case, as each chemical sensor within the array is different, its interaction with the sample is unique, allowing the pattern of responses to be learnt by a pattern-recognition technique. Though successful, these instruments are prone to be affected by drift and environmental factors and have a relatively poor sensitivity [14]. More recently, techniques such as FAIMS and drift-tube ion mobility spectrometry (IMS) have found favour due to their higher sensitivity and reduced drift [15]. These techniques have similar properties to electronic noses, allowing for simple operation, typically not requiring specialised carrier gases, and a lower cost point per sample. In addition, standard IMS has now been complemented with a pre-separation gas chromatography (GC) technique to help increase the information content in a complex headspace [16]. This is advantageous over standard FAIMS techniques, which are based solely on the mobility of ions in varying electric fields. In IBD diagnosis, FAIMS and IMS have been successfully implemented to analyse VOCs through utilising a variety of sample types, including human serum, urine, breath, and faecal matter [10,17,18].

In this study, we therefore aimed to compare the VOC profiles of de novo therapy-naïve paediatric IBD patients with those of CGIs and to compare the VOC profiles of IBD subtypes using GC-IMS. Due to its multifactorial aetiology and complex pathogenesis, paediatric IBD is a heterogeneous disease with regards to disease localisation, extent, and severity. Current guidelines recommend initiating induction therapy with the least-intensive treatment options (step-up approach), unless severe disease manifestations such as bowel strictures or fistulas are present at baseline [19,20]. However, not all patients benefit from this approach and may require more aggressive medications sooner to improve disease outcome and quality of life [21]. Personalised medicine tools for the prediction of response to therapy at baseline are currently lacking. A secondary aim of our study was, therefore, to investigate if baseline VOC profiles allow for predicting response to induction treatment in paediatric IBD patients at baseline.

## 2. Materials and Methods

### 2.1. Study Design

Patients aged 4–17 years with gastrointestinal symptoms and referred to the outpatient clinics of the Department of Paediatric Gastroenterology at Amsterdam University Medical Centre, both location AMC and VUmc, between December 2017 and June 2023 were eligible for inclusion. Exclusion criteria were the use of antibiotics or probiotics during the three months prior to inclusion, the use of immunosuppressant therapy prior to inclusion, and diagnosis of an immunocompromised disease. Some patients were referred with suspicion of IBD and underwent a complete diagnostic IBD work-up according to current guidelines, including clinical assessment, laboratory tests, radiological imaging, and endoscopy with histology [22]. Others were referred based on suspicion of functional abdominal pain (FAP) or irritable bowel syndrome (IBS) and underwent clinical assessment and laboratory tests if indicated. 

Clinical data were collected from the electronic patient files, including medical history, physical examination, laboratory results, radiological findings, endoscopic findings, histological results, final diagnosis, Paediatric Crohn’s Disease Activity Index (PCDAI), Paediatric Ulcerative Colitis Activity Index (PUCAI), and choice of induction treatment. For patients diagnosed with IBD, data regarding the response at three months after induction of therapy were collected. Response was defined as a combination of clinical response, defined as a decrease in PCDAI of ≥12 points in the case of CD or a decrease in PUCAI of ≥20 points in the case of UC or IBD-U, and not requiring treatment escalation to a corticosteroid or biological agent within the first three months of therapy.

All patients were instructed to collect a stool sample at home using a provided sterile container. If applicable, this was performed before bowel cleansing. Subsequently, they were required to store the sample in their home freezer and bring it under cooled conditions to their next hospital visit. The sample was then stored at −80 °C until further processing.

### 2.2. Sample Analysis

VOCs were analysed using GC-IMS (FlavourSpec^®^, G.A.S., Dortmund, Germany). GC-IMS is an analytical technique able to separate chemical compounds based on both their retention times in a GC column and their drift times in IMS. Samples were prepared by transferring a 0.5 g faecal sample to a 20 mL air-tight headspace vial with a magnetic cap. Then, each sample was heated to 40 °C and agitated for 10 min prior to sampling. Once completed, the headspace from the sample was injected into the GC-IMS equipment for analysis. During the analysis, the VOCs in the headspace were initially separated by the GC column and then subsequently chemically ionised with a tritium source. The ionised VOCs were then pulled through the drift tube with a high electrical field while simultaneously buffered with a drift gas, in this case nitrogen. This allows for the VOCs to be separated by their size-to-charge ratio, with larger molecules drifting slower than small molecules, resulting in variable drift times. GC-IMS produces high-dimensionality spectrums, containing drift time, the ion’s intensity level, and retention time [16]. Further information on sample preparation and GC-IMS parameters have been previously described in the literature [23,24].

### 2.3. Statistical Analysis

Demographic and clinical data were analysed using the Statistical Package for Social Sciences (SPSS, IBM; v28). Normally distributed data were shown as means and standard deviations (SD), whereas non-normally distributed data were depicted as medians and interquartile ranges (IQR). Continuous variables were compared using the non-parametric Mann–Whitney U test. A *p*-value < 0.05 was considered statistically significant. 

The high dimensionality data generated by the GC-IMS technique contain more than 10 million data points per sample. However, chemical information is relatively sparse and is typically localised to a small region of the dataset. Therefore, pre-processing steps are introduced to reduce this dimensionality, which are further outlined in a previously described protocol [25]. In summary, the data are visually inspected and the region containing all the identified chemical information is then cropped out of the wider dataset, with the same settings used across all the samples. Next, the reactive ion peak (RIP) is removed by selecting a line across the spectra (drift time) that contains the RIP but excludes any chemical information. This line then subtracted from the rest of the data for that sample. In addition, a small static threshold is added to remove the background noise for the entire dataset, where values below this threshold are set to zero. The line subtracted from each file and the magnitude of the threshold is the same for all samples/files. This uses a custom program written in LabVIEW 2023 (National Instruments, Austin, TX, USA). 

Once the data underwent pre-processing, the data were then analysed using a custom feature selection pipeline in R (R Foundation for Statistical Computing, Vienna, Austria; version 2022·07·1—Build 554). This method performs a binary class prediction using a k-fold cross-validation methodology that separates the dataset into ten equally sized subsets (10-fold cross validation). Thereafter, a single subset was used for validation, and the other nine were used to train the data. This was repeated until all the data were used as a test sample. 

To identify features (data points) that hold discriminatory information, a rank-sum test was undertaken and features with the lowest *p*-values were selected from training set. This process was only conducted with the training set to reduce potential data leakage. It is important to note that the identified features are obtained solely on a statistical basis and do not have any biological significance. Thereafter, the top features were implemented into sparce logistic regression and XGboost classifier algorithms. These were chosen as they are well-established algorithms and have been used successfully in previous datasets. Sparce logistic regression (R Package: glmnet—version 4.1-6) is good at dealing with high-dimensional datasets, whilst XGBoost (R package: XGboost version—1.6.0.1) is a method that assembles multiple models (usually decision trees) to create a strong prediction [26,27]. From the resultant probabilities, statistical values were calculated, including the receiver operating characteristic (ROC) curve, the area under the curve (AUC), sensitivity, specificity, the positive predictive value (PPV), the negative predictive value (NPV), and *p*-values. These analyses were computed using R packages.

### 2.4. Sample Size Calculation

Based on the results of a previous, smaller study of our group on VOC profiles in paediatric IBD and IBS patients, we calculated the sample size [14]. To assess the ability of VOCs to distinguish IBD from controls, a random sample size of 123 paediatric IBD subjects and 62 non-IBD controls would produce a two-sided 95% confidence interval (CI) with a width of 0.10 when the sample AUC is 0.87.

## 3. Results

### 3.1. Patient Characteristics

We included 109 de novo therapy-naïve IBD (74 CD, 29 UC, and 6 IBD-U) patients and 75 CGIs. Patient characteristics of the study population are depicted in Table 1. The final diagnoses of the CGIs are shown in Table 2. 

There were no significant differences in age, sex, and BMI between the IBD group and the CGI group. FCP and CRP levels were significantly higher in IBD patients compared to CGIs (both *p* < 0.001). There were no significant differences in age, sex, BMI, FCP, and CRP between IBD phenotypes. Follow-up data at three months after the initiation of induction therapy of 72 IBD patients were available for response analysis. Of these 72 subjects, 34 were categorised as responders, and 38 were classified as non-responders. At baseline, there were no significant differences in age, sex, BMI, FCP, CRP, PCDAI, or PUCAI between responders and non-responders.

### 3.2. Faecal Volatile Organic Compound Analysis

Figure 1 depicts a standard GC-IMS output of an IBD subject. As can be observed, the background is represented in dark blue, with the red and lighter blue areas indicating detected molecules. It can be observed that most of the detected molecules are concentrated in the bottom-left of the plot, thus providing the capacity to crop the data without removing any chemical information. A prominent feature in an IMS output is the red line, known as the RIP. It is produced from the continuous ionisation of the carrier gas, which forms reactant ions capable of inducing a chemical reaction with the analytes. The peak intensities (highest represented in red) are indicative of the number of ions and, therefore, a direct link to a chemical compound. Finally, the figure shows there was minimal carryover between chemicals.

The GC-IMS results of the VOC analysis are depicted in Table 3. For each comparison, results from both classifier algorithms are presented. IBD patients could significantly be discriminated from CGIs with a moderate diagnostic accuracy (AUC ± 95% CI, sensitivity, specificity, PPV, NPV, *p*-values: 0.71 (0.64–0.79), 0.59, 0.77, 0.63, 0.73, <0.001). Similarly, CD patients showed significant differences in VOC profiles from CGIs (0.75 (0.67–0.84), 0.72, 0.72, 0.77, 0.67, <0.001), and UC patients could also significantly be distinguished from CGIs (0.67 (0.56–0.78), 0.63, 0.68, 0.82, 0.43, 0.01). There was no significant difference between the CD and UC patients (0.57 (0.45–0.69), 0.42, 0.7, 0.39, 0.75, 0.87). Responders could not significantly be distinguished from non-responders (0.70 (0.58–0.83), 0.78, 0.54, 0.60, 0.74, 0.1). ROC curves for these tests are shown in Figure 2.

## 4. Discussion

In this study, we found that faecal VOC patterns of paediatric IBD patients are significantly different from those of paediatric CGIs. IBD subtypes also showed significant differences in VOC composition when compared to CGIs, whereas IBD subtypes could not significantly be distinguished from each other. Similarly, responders did not show significant differences in VOC profiles than non-responders.

This is, to our knowledge, the first case–control study comparing VOC compositions of de novo therapy-naïve paediatric IBD patients with those of CGIs using GC-IMS. Previously, we demonstrated differences in VOC patterns of paediatric IBD patients and those of IBS/FAP patients through FAIMS [11]. Though the gold standard for chemical analysis is GC-MS, it requires specialised laboratory facilities and dedicated staff [12]. Both FAIMS and GC-IMS are lower-cost platforms with high sensitivity and can use air or nitrogen as the carrier gas. GC-IMS has the advantage of some level of pre-separation using a GC technique. In our case, GC-IMS has the further advantage of being fitted with an autosampler, allowing for controlled sample introduction. However, the disadvantage of our study is that we did not undertake chemical identification of the headspace components, due to resource limitations. Nonetheless, with the results of the current study we have validated our previous findings that faecal VOC composition differs between paediatric IBD patients and CGIs with a more sensitive and suitable technique.

Comparable results have previously been found in adults. In a cohort of 30 IBS patients and 110 IBD patients, Ahmed and colleagues showed that VOCs could discriminate IBS patients from IBD patients with an AUC of 0.98 [8]. They identified 60 significantly differentiating VOCs, with 50 VOCs showing higher levels in IBS, while the remaining 10 VOCs exhibited greater abundance in IBD. Among the former were predominantly short-chain fatty acids and cyclohexanecarboxylic acid and its derivatives, while the latter consisted mainly of compounds from the aldehyde class. Aldehydes are associated with oxidative stress and have often been found in higher abundance in various matrices in patients with various types of inflammatory diseases [8,28,29]. Recently, a systematic review by Zhang and colleagues similarly showed that IBS patients can significantly be distinguished from both IBD patients and healthy controls through VOC analysis [30]. 

We did not find a significant difference in the VOC composition between IBD patients who responded well to induction treatment and patients who did not. To our knowledge, ours is the first study to investigate the potential of VOCs to predict the response to therapy in paediatric IBD patients. In adults, comparable studies have been conducted. Rossi et al. found that VOCs were capable of predicting the response to dietary interventions in a group of 93 adult IBS patients [31]. Another study from our research group in adult IBD patients found that VOCs were able to predict disease course by seeing alterations in VOC composition prior to a change in FCP levels [32]. The findings of these studies suggest the potential of VOCs to predict treatment response and disease progression. However, our present study does not demonstrate such predictive capability. This could be explained by the fact that children and adults exhibit different IBD phenotypes [33,34]. Additionally, possibly due to the limited sample sizes across all three studies, the chemical concentrations were below the limit of detection of our system, or the proton affinity was low for these chemicals. Nonetheless, further investigation into these associations remain warranted, as it could provide opportunities for non-invasive monitoring and predicting disease course in a personalised manner.

A strength of our study is the inclusion of de novo therapy-naïve paediatric IBD patients and the use of strict exclusion criteria, minimising the influence of antibiotics, other medications, or IBD therapy on VOC profiles. Another strength is the use of GC-IMS, a highly sensitive technique for measuring VOCs in a complex matrix such as faeces. A limitation of our study is that this methodology only allows for VOC pattern identification, without providing the specific VOCs that are altered between the groups. This complicates drawing conclusions regarding the translation from VOC outcome to increasing the understanding of the pathophysiology, though it does lay a solid foundation for further research by demonstrating associations. Furthermore, in this study we included patients with a variety of non-IBD diagnoses, of which IBS and FAP were most common, creating a heterogeneous control group. Some of these alternative diagnoses are known to have altered faecal VOC profiles, such as coeliac disease [35]. However, it is crucial to note that the patient population referred due to gastrointestinal symptoms is also heterogeneous and requires accurate differentiation from IBD in clinical practice. Moreover, we did not reach our pre-calculated sample size of 123 paediatric IBD subjects and 62 non-IBD controls due to our prospective study design, since some patients initially referred under suspicion of IBD did not meet the diagnostic criteria and were consequently classified as CGIs. We do not believe this influences our primary outcomes, as we ultimately reached the same total sample size. However, it did lead to a smaller IBD group size, which may have affected our ability to compare response data. Additionally, cases and controls were not formally matched on possible confounders such as age, sex, and BMI, though these factors did not significantly differ between the groups at baseline, so possibly have not influenced the outcome. Moreover, we have not accounted for the potential influence of diet and other environmental factors other than medication use on faecal VOC composition, possibly affecting the overall outcome [36,37]. Since subjects from both groups derive from the same regions in The Netherlands, it can be speculated that there are no differences in dietary habits and other environmental factors, limiting the risk of type I error.

## 5. Conclusions

In this study, we have demonstrated that baseline VOC compositions as measured by GC-IMS differ between de novo paediatric IBD patients and CGIs. Secondly, baseline VOC profiles were not able to predict treatment response in paediatric IBD patients. 

The discriminatory ability of VOCs holds significant importance given clinicians’ frequent need to differentiate between patients with IBD and those with gastrointestinal symptoms but without IBD. Understanding the distinctions in VOC profiles among individuals with IBD and CGIs is crucial for translating and implementing VOC research findings into clinical practice. Future research on VOC analysis should focus on the additional value of VOCs in the diagnostic work-up of paediatric IBD, as currently used biomarkers such as FCP have limited specificity.

## Figures and Tables

**Figure 1 sensors-24-02727-f001:**
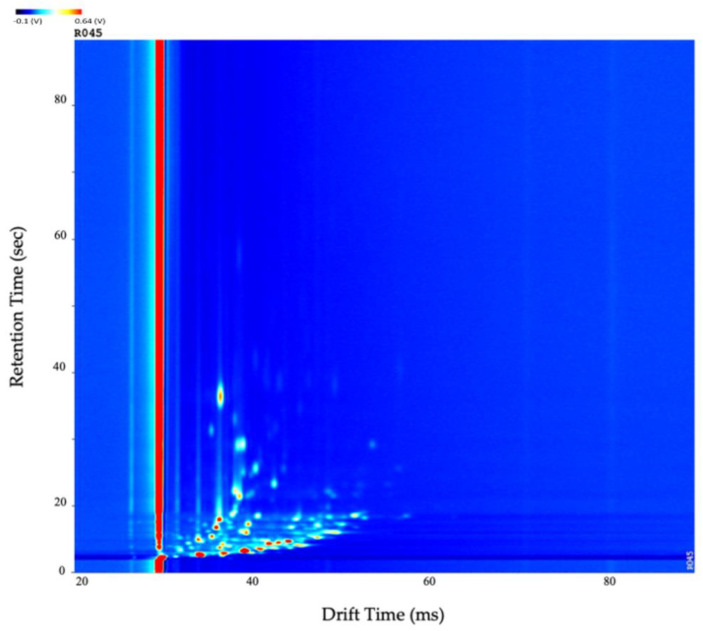
The standard topographic plot obtained in an IBD subject with the drift and retention time normalised (0–100). Abbreviations: IBD, inflammatory bowel disease.

**Figure 2 sensors-24-02727-f002:**
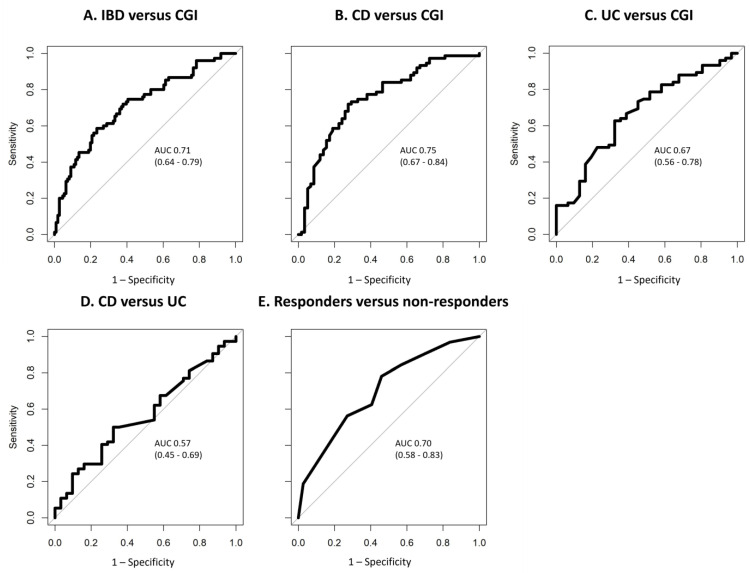
ROC curves of VOC profiles for the discrimination of (**A**) IBD versus CGI; (**B**) CD versus CGI; (**C**) UC versus CGI; (**D**) CD versus UC; (**E**) responders versus non-responders. The ROC curves of the best-performing classification algorithms are shown, and AUC and 95% CI are reported. Abbreviations: IBD, inflammatory bowel disease; CGI, control with gastrointestinal symptoms; CD, Crohn’s disease; UC, ulcerative colitis; ROC, receiver operating characteristic; VOC, volatile organic compound; AUC, area under the curve; 95% CI, 95% confidence interval.

**Table 1 sensors-24-02727-t001:** Patient characteristics. Abbreviations: IBD, inflammatory bowel disease; CD, Crohn’s disease; UC, ulcerative colitis; IBD-U, inflammatory bowel disease—unclassified; CGI, controls with gastrointestinal symptoms; IQR, interquartile range; BMI, body mass index; PCDAI, Paediatric Crohn’s Disease Activity Index; PUCAI, Paediatric Ulcerative Colitis Activity Index; FCP, faecal calprotectin; CRP, C-reactive protein; NA, not applicable.

	IBD (*n* = 109)	CD (*n* = 74)	UC (*n* = 29)	IBD-U (*n* = 6)	CGI (*n* = 75)
Age, years, *median* (*IQR*)	15.0 (12.5–16.0)	15.0 (13.0–16.0)	14.0 (12.0–15.5)	13.5 (11.7–15.3)	14.0 (11.0–16.0)
Males, *n* (*%*)	54.0 (49.5)	37.0 (50.0)	14.0 (48.3)	3.0 (50.0)	34.0 (45.3)
BMI, kg/m^2^, *median* (*IQR*)	18.9 (16.0–21.6)	18.9 (15.7–21.3)	18.5 (16.8–21.8)	18.7 (16.8–22.2)	19.1 (17.0–21.1)
PCDAI baseline, *median* (*IQR*)	NA	30 (21.9–42.5)	NA	NA	NA
PUCAI baseline, *median* (*IQR*)	NA	NA	45 (30–65)	30 (23.75–36.25)	NA
FCP, μg/mg, *median* (*IQR*)	1987.5 (1027.0–3267.5)	1892.5 (886.2–3000.0)	2170.5 (1270.0–4734.0)	2560.5 (1112.7–3551.0)	25.5 (11.0–317.8)
CRP, mg/L, *median* (*IQR*)	8.2 (1.0–32.9)	12.0 (5.5–48.9)	1.0 (0.7–8.0)	0.9 (0.6–2.8)	1.0 (0.3–3.7)

**Table 2 sensors-24-02727-t002:** Final diagnoses of the control group. Abbreviations: IBS, irritable bowel syndrome; FAP, functional abdominal pain.

Final Diagnosis	Patients, *n* (%)
IBS	34 (45.3)
FAP	18 (24.0)
Polyp	6 (8.0)
Post-infectious	5 (6.7)
No alternative diagnosis (transient symptoms)	3 (4.0)
Non-specific colitis (non-IBD)	2 (2.7)
Abdominal migraine	1 (1.3)
Chronic appendicitis	1 (1.3)
Coeliac disease	1 (1.3)
Gynaecological disease	1 (1.3)
*Helicobacter pylori* gastritis	1 (1.3)
Protein-losing enteropathy	1 (1.3)
Ulcer seam (after prior intestinal resection)	1 (1.3)

**Table 3 sensors-24-02727-t003:** Differences in faecal VOC patterns. AUC, sensitivities, specificities, PPV, NPV, and *p*-values are reported for the respective optimum cut-off points. Abbreviations: IBD, inflammatory bowel disease; CD, Crohn’s disease; UC, ulcerative colitis; VOC, volatile organic compound; AUC, area under the curve; 95% CI, 95% confidence interval; PPV, positive predictive value; NPV, negative predictive value.

Comparison	Algorithm	AUC (95% CI)	Sensitivity	Specificity	PPV	NPV	*p*-Values
IBD versus controls	Sparse logistic regression	0.71 (0.64–0.79)	0.59	0.77	0.63	0.73	<0.001
	XGboost	0.68 (0.60–0.75)	0.49	0.79	0.62	0.70	<0.001
CD versus controls	Sparse logistic regression	0.74 (0.65–0.83)	0.93	0.52	0.71	0.86	<0.001
	XGboost	0.75 (0.67–0.84)	0.72	0.72	0.77	0.67	<0.001
UC versus controls	Sparse logistic regression	0.66 (0.54–0.77)	0.65	0.74	0.86	0.47	<0.01
	XGboost	0.67 (0.56–0.78)	0.63	0.68	0.82	0.43	0.01
CD versus UC	Sparse logistic regression	0.57 (0.45–0.69)	0.42	0.73	0.39	0.75	0.87
	XGboost	0.54 (0.42–0.66)	0.18	0.94	0.87	0.32	0.28
Responder versus non-responder	Sparse logistic regression	0.61 (0.47–0.75)	0.37	0.95	0.86	0.64	0.06
	XGboost	0.70 (0.58–0.83)	0.78	0.54	0.56	0.74	0.1

## Data Availability

The raw data will be made available on request, as they are linked with our ethical approval.

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
