# Peer review of "Faecal Volatile Organic Compound Analysis in De Novo Paediatric Inflammatory Bowel Disease by Gas Chromatography–Ion Mobility Spectrometry: A Case–Control Study"

_sensors, 2024, doi:10.3390/s24092727_

Round 1
Reviewer 1 Report
Comments and Suggestions for Authors
The manuscript gives a comprehensive and well written description of the study carried out. Even the major drawback of the study - not having identified the relevant peaks and the particular VOCs behind - is mentioned appropriately. Therefore, I recommend publication and I have only few minor comments (see below).
Line 71-73: FAIMS is not even written out, but the reference is on FAIMS only, please extend.
Line 148-150: This sentence describes indeed removal of the noise but not of the background and the RIP, please give a little more details.
Table 2: Use the same number of decimals in line “age” and “male”.
Reviewer 2 Report
Comments and Suggestions for Authors
Overall, this research highlights the importance of accurately discriminating VOCs for clinicians in differentiating patients with IBD from those with similar gastrointestinal symptoms. Further exploration of VOC analysis, particularly in pediatric IBD cases, is essential to enhance diagnostic precision beyond the limitations set by current biomarkers. While this study has made a valuable contribution to the field, there are certain aspects that require further elaboration:
1.In lines 137-141 of the manuscript’s main text, the statistical methods used appear appropriate, but it would be helpful to include more details on the statistical tests performed and the rationale for their selection.
2.This study introduced a clinical guidance instrument analysis method and recommends conducting methodological investigations to validate its accuracy and stability of the analytical approach.
3.As a means of clinical examination, it is recommended that the authors use the method established in the article for practical validation to demonstrate the utility of the method.
4.In lines 266-279 of the manuscript’s main text, the authors highlighted a limitation in the experimental findings: “No significant difference was observed in VOC composition between responding and non-responding IBD patients.” It is suggested that the authors elaborate on this discrepancy by citing pertinent literature.
5. In conclusion, this study has the potential to make a significant contribution to the understanding of inflammatory bowel disease in paediatric patients. Addressing the above-mentioned points will strengthen the quality and impact of the paper. I recommend revisions based on these feedback points before considering publication.
